## Research Article

suicidal behaviour; suicide attempt; self-harm; mental health; young adolescents; parental absence

**Corresponding author:**
Joemer Calderon Maravilla;
Email: j.maravilla@uq.edu.au

# Suicidal behaviours and self-harm among adolescents: Results from a school-based mental health survey in the Philippines

Daisy Huelva Alberto[1], Restituta C. Tan[1], Jonathan P. Guevarra[2], Alely S. Reyes[1], Irma Peñalba[3], Anne Abio[4,5], Andre Sourander[4,5,6] and Joemer Calderon Maravilla[7,8,9] 

[1]College of Nursing, De La Salle Medical and Health Sciences Institute, Philippines; [2]College of Public Health, University of the Philippines, Manila, Philippines; [3]Special Health Sciences Senior High School, De La Salle Medical and Health Sciences Institute, Dasmariñas, Philippines; [4]Research Centre for Child Psychiatry, University of Turku, Finland; [5]INVEST Research Flagship Center, University of Turku, Turku, Finland; [6]Department of Child Psychiatry, Turku University Hospital, Finland; [7]School of Public Health, The University of Queensland, Herston, Australia; [8]Queensland Centre for Mental Health Research, Wacol, Australia and [9]Institute of Health Sciences and Nursing, Far Eastern University, Manila, Philippines

## Abstract

There is limited post-pandemic youth mental health data in low- and middle-income countries. This study describes the prevalence of suicidal ideation, suicidal attempt, and self-harm since the COVID-19 pandemic among young Filipino adolescents. Adolescents aged 13-16 years old from public and private high schools in Cavite, Philippines were recruited for a cross-sectional school survey conducted from May 2023 to February 2024. Suicidal behaviours and self-harm since the pandemic were determined using a self-administered questionnaire alongside sociodemographics and internalising and externalising symptoms. Of the 1,229 13-16-year-olds who completed the survey, 54.0% experienced suicidal ideation, 24.2% attempted suicide, and 34.2 % reported self-harm between 30 January 2020 and the date when they completed the survey. The prevalence of suicide attempts was higher among females (29.6%) than males (13.1%). Parental absence was associated with suicidal attempts (ARRR=2.93) and self-harm and/or suicidal ideation (ARRR=2.00) while living with either the biological mother or father was moderated by gender. Internalising and externalising symptom scores increased the risk for both outcomes by ≥15%. This study revealed a high prevalence of suicidal and self-harming behaviours among young adolescents in the Philippines. This calls for action to implement population-based strategies in suicide prevention, early screening, and cross-sectoral intervention.

## Impact statements

This study provides critical data on the mental health problems experienced by young Filipino adolescents during the tail-end of the coronavirus disease 2019 pandemic. This study revealed a high prevalence of suicidal ideation, suicide attempts and self-harm among adolescents aged 13–16 years. Notably, one in four adolescents reported attempting suicide between 30 January 2020 and the date when they completed the survey, while one in three had either engaged in self-harm or considered attempting suicide. Beyond internalising and externalising symptoms, parental absence emerged as a significant risk factor for suicidal ideation, suicide attempts and self-harm, particularly among girls. These intersecting vulnerabilities highlight the urgent need for targeted mental health interventions. The study underscores the importance of implementing youth mental health screening and early intervention programs in both school and community settings. In light of the rapidly evolving social norms and environments post-pandemic, it is essential to promote parental involvement in clinical care and adopt holistic public health strategies for suicide prevention.

## Introduction

Suicidal behaviours and self-harm are major public health concerns due to their significant contribution to the global burden of disease and premature mortality. Although these behaviours often coexist and share common predictors (Grandclerc et al., 2016), self-harm is conceptually distinct from suicidal behaviour as self-harm refers to self-inflicted pain or injury without the intent to die (Australian Institute of Health and Welfare, 2025). Globally, suicide accounts for ~726,000 deaths across all age groups (World Health Organization, 2024). It is the third leading

cause of death among 15- to 29-year-olds, with 8,327 reported deaths among adolescents aged 10–14 years in 2019 (Michalek et al., 2024). Between 2000 and 2022, self-harm affected 17.7% of adolescents aged 10–19 years (Denton and Álvarez, 2024), with higher prevalence among girls than boys (Moloney et al., 2024).

Recent global studies have reported a 25% increase in the prevalence of mental disorders and suicidal behaviours following the coronavirus disease 2019 (COVID-19) pandemic, with adolescents more likely than adults to experience symptoms of anxiety and depression (Santomauro et al., 2021; World Health Organization, 2022). These mental health problems were exacerbated by intermittent school closures, which disrupted adolescents' social development and daily routines (Cornish and Smart, 2023). Despite growing evidence on the prevalence of suicidal ideation, suicide attempts and self-harm among adolescents, data remains limited during and after the pandemic, particularly in low- and middle-income countries (LMICs). For instance, recent analyses of the Global School Health Survey (GSHS), a nationally representative survey targeting 13- to 17-year-olds in LMICs, have solely relied on data collected in 2019. This leaves a significant gap in understanding adolescent mental health in the post-pandemic context (Li et al., 2021). Moreover, a global review examining suicide attempts, suicide mortality and self-harm during the pandemic across 12 LMICs identified 22 studies, most of which used facility-level records rather than community-based samples with data coverage only up to October 2020, potentially introducing bias and limiting generalisability (Knipe et al., 2022).

In the Association of Southeast Asian Nations region, the Philippines reported the highest pre-pandemic past-year prevalence of suicide attempts among adolescents aged 12–15 years in 2015, followed by Thailand and Malaysia (Pengpid and Peltzer, 2020; Li et al., 2021). The Young Adult Fertility and Sexuality Study (YAFS), a nationally representative youth survey in the Philippines, revealed a twofold increase in the lifetime prevalence of suicide attempts among 15- to 24-year-olds from 3.0% in 2002 to 7.5% in 2021 (University of the Philippines Population Institute, 2022). However, YAFS did not include young people in early adolescence (i.e., 10–14 years), a critical period in the development of common adolescent mental disorders (McGrath et al., 2023). In terms of suicide mortality, 10- to 14-year-olds in the Philippines have shown increasing trends since 2015, contrasting with declining rates among 15- to 24-year-olds (Maligalig, 2021). Despite these trends, links between suicidal behaviours, self-harm and youth psychopathology remain underexplored in the Philippines. Most local studies have focused on internalising symptoms, such as depression and anxiety, with limited investigation into externalising symptoms, such as conduct problems, hyperactivity and inattention (Renaud et al., 2022).

In 2024, the Philippines has just recently enacted the "Basic Education Mental Health and Well-Being Promotion Act" to support the delivery of mental health programs and services for young people in schools (Republic of the Philippines, 2024). Updated and comprehensive mental health data are essential to guide health investments and tailor support for young people at risk of self-harm and suicide attempts. While most available data focus on older adolescents, there is a critical need to understand the mental health needs of younger adolescents, who often face unique challenges related to biological and emotional development, as well as psychosocial transitions, such as moving from primary to secondary school. More broadly, in LMICs, research involving younger adolescents is vital to inform public health strategies for school-based mental health promotion and community-level interventions, especially in contexts where access to specialist services remains limited.

This study aims to describe the prevalence of suicidal behaviours and self-harm among Filipino adolescents aged 13–16 years. Specifically, it examines experiences of suicidal ideation, suicide attempts and self-harm since January 2020, marking the onset of the COVID-19 pandemic. The study also explores prevalence patterns across gender, age, grade level, socio-economic status and type of school. In addition, it investigates associations between suicidal behaviours, self-harm and indicators of social and emotional well-being.

## Methods

### Study design

A cross-sectional survey was conducted in public and private high schools in two selected cities in Cavite, Philippines: Dasmarinas City and Trece Martirez City. This study was reviewed and approved by the De La Salle Medical and Health Sciences Institute Independent Ethics Committee (approval number: GSER-038), which adheres to the 2022 National Ethics Guidelines for research involving human participants in the Philippines.

### Study setting

Cavite was chosen as the study site due to its status as the most populous province in the Philippines, with a total population of 4.34 million in 2020 (Mapa, 2021; Provincial Government of Cavite 2022). The province has undergone rapid industrialisation, attracting a diverse population of migrants and relocated informal settlers from Metro Manila (or the National Capital Region). In 2023, Cavite recorded the highest poverty threshold and the highest number of individuals in poverty compared to its surrounding provinces (Philippine Statistics Authority Region IVA – CALABARZON, 2025). Among Cavite's 16 municipalities and seven cities, Dasmarinas City has the highest population, accounting for 16.2% (703,141) of Cavite's total population in 2020, and hosts the greatest number of educational institutions (16 public and 94 private) in the province (Mapa, 2021; Provincial Government of Cavite 2022). Trece Martirez City, the provincial capital, had a population of more than 200,000 (0.5%) in 2020 (Trece Martirez City Government, 2023). The survey was strategically implemented in these cities due to their high proportion of young people aged 10–19 years, as indicated by the 2020 Philippine Census, and an increase in poverty incidence since the pandemic (Philippines Statistics Authority, 2024; Provincial Government of Cavite 2022).

### Sampling

High school students aged 13–16 years old, enrolled from May 2023 to February 2024, were recruited to participate in the survey using non-probability sampling. We anticipated a sample size of 1,000 students, which provides 99.9% statistical power (i.e., <1% probability of Type II error) at a 0.05 significance level to detect a suicide attempt prevalence of 7.5%, as reported in the 2021 YAFS (University of the Philippines Population Institute, 2022).

Approvals from the gatekeepers, including the jurisdictional education department at the city and provincial levels and respective school principals, were obtained. A total of 15 high schools (8 public schools and 7 private schools) agreed to be engaged in the survey. Written consent was sought from both parents and adolescents before survey participation. Consent from parents was

obtained by sending them an invitational letter, the information sheet and the consent form. These were sent 1 week before the data collection in parallel with information sessions online or on-site, depending on the preference of the participating school. These sessions provided parents with the opportunity to ask questions about the survey. Only the adolescents with signed consent from parents were eligible to participate and were provided with a study information sheet and assent form (paper-based or electronic) on the day of the survey administration. Before the administration of the survey on the day, adolescents were also given the opportunity to raise concerns and questions, as well as withdraw from the study. Non-consenting adolescents were allowed to withdraw despite parental consent.

## Procedures

Participating schools provided designated rooms where adolescents could complete the questionnaire. Although the survey was self-administered in group settings, adequate spacing was ensured to maintain individual privacy. At least one member of the research team was present in each room during data collection to oversee the process and provide support to adolescents if needed. Teachers were permitted to remain in the room to provide support in case any adolescents experienced distress and to advocate on behalf of adolescents who had questions or concerns about the study. However, teachers were not allowed to view adolescents' responses or assist in interpreting questionnaire items.

Adolescents were offered the option to self-complete the paper-based or the electronic version (via Qualtrics) of the questionnaire; 76.9% (n = 945) of the adolescents chose the paper version. Upon receipt, completed paper-based questionnaires are immediately placed in an envelope and securely stored in a locked cabinet accessible only to DHA (i.e., the first author). For students who completed the survey electronically, their responses were transmitted directly to Qualtrics. To ensure confidentiality, the questionnaire did not collect students' names or any personally identifiable information. The average survey time among students who used a paper-based questionnaire was 30 min, while the average survey time among those who used the electronic questionnaire was 35 min.

To mitigate distress, adolescents were encouraged to take at least a one-min break during the survey; prompts were also built into the questionnaire. A distress protocol was implemented in cases where participants reported that they were in distress or when the researcher observed signs of distress. Those who experienced distress were initially provided with support by our team, who are trained counsellors (DHA, RCT and ASR). Distressed participants were offered to be referred to their respective school counsellor, when available, or preferred teacher-confidante for further management. A total of seven students experienced distress during the administration of the survey. Only one needed further referral to their teacher-confidante due to recall of their own family-related issues. A follow-up call was made to the teacher-confidante, who reported no further signs of distress. All participants were provided with a list of available support services following the survey. No participants were reported to be in distress after completing the survey.

## Study measures

The questionnaire is composed of 10 modules and 125 items. The nature of the items in each module was explained in Supplementary Table S1. The questionnaire was originally written in English. All items except the Strengths and Difficulties Questionnaire (SDQ) were translated into Filipino and back-translated in consultation with experts from the Language Department of the De La Salle Medical and Health Sciences Institute. The English and Filipino versions of the questionnaire underwent face validity and pilot testing. While both versions were made available, all adolescents used the English version.

This current research used the following measures.

### Demographic factors

Demographic characteristics considered in this study include gender (female, male and other), age in years, grade (or year level), type of school (public, private), perceived socio-economic status of the family (not well, not particularly well, fairly well, rather well and very well), living arrangements (living with biological mother, father, both or neither).

### Suicidal ideation, suicidal attempt and self-harm

Suicidal ideation, suicidal attempts and self-harm behaviours were assessed using a recall period between the onset of the COVID-19 pandemic, defined as 30 January 2020, and the date when the adolescents completed the survey (i.e., May 2023 to February 2024). Suicidal ideation was asked using the question, "Since 30 January 2020, have you seriously thought about committing suicide?" ("*Mula noong 30 January 2020, naisip mo na bang magpakamatay?*"). Adolescents were then given the following response options: no, yes (once) and yes (more than once). With the same response options, suicide attempt during the pandemic was asked using the question, "Since 30 January 2020, have you tried committing suicide?" ("*Mula noong 30 January 2020, sinubukan mo na bang magpakamatay?*"). Self-harm was assessed using the question, "Since 30 January 2020, have you intentionally hurt yourself, for example, by cutting or burning your skin?" ("*Mula noong 30 January 2020, sinaktan mo na ba ang iyong sarili tulad ng paglalaslas at pagsunog ng balat?*"). Adolescents were allowed to answer "No," "Yes (once)" or "Yes (more than once)". Although these questions (in both English and Filipino) were adapted from the 2019 Philippines GSHS, we further assessed adolescents' understanding of these questions during pilot testing to ensure their validity.

### Socio-emotional well-being

The SDQ was administered to adolescents to assess their socio-emotional well-being. The SDQ is developed by Goodman (2001) and is composed of 25 items, which can generate a total score ranging from 0 to 40, as well as scores for three subscales: internalising symptoms (e.g., emotional and peer problems), externalising symptoms (e.g., hyperactivity and conduct problems) and prosocial behaviours (e.g., positive social skills). Higher internalising and externalising scores indicate poor mental health, while higher prosocial scores indicate positive mental health.

Because the youth SDQ is not available in the Filipino language, the English version of the SDQ was provided to adolescents who opted to use the Filipino version of the questionnaire. SDQ is known to have adequate psychometric properties to assess psychopathologies across cultures, including an acceptable internal validity (Cronbach's $\alpha$ = 0.80) (Goodman, 2001) and a stable construct validity of the three established subscales (Sourander et al., 2024).

### Statistical analysis

The majority of the analyses were conducted using Stata18 (StataCorp, 2023). Descriptive statistics, including frequency distribution, means and interquartile range, were examined. An upset plot was generated to determine intersections between suicidal ideation, suicide attempt and self-harm using R Studio (Conway

and Gehlenborg, 2019). The proportions of those who experienced suicidal ideation, suicidal attempt and self-harm were calculated by dividing the number of adolescents who answered yes (once) or yes (more than once) by the total sample.

The association between socio-demographic characteristics, suicidal ideation, suicide attempt and self-harm was determined using multinomial logistic regression. To facilitate the regression analysis, the outcome assessed included three categories: (1) those who did not experience suicidal ideation, suicidal attempt or self-harm as the base outcome; (2) those who experienced suicidal ideation and/or self-harm but not suicidal attempt; and (3) those who experienced suicidal attempt. The model was adjusted for age, perceived family economic status, biological parents living with the adolescent, type of school, SDQ scores and intragroup correlation by school. All participants provided complete information on all measures, except for four in SDQ; these were excluded from the regression analysis.

Effect modification by gender was investigated through the inclusion of interaction terms between gender and covariates (i.e., gender, age, perceived family economic status, biological parents living with the adolescent, type of school and SDQ scores). The interaction models omitted those who identified as the other gender from the analysis due to small cell size. The multivariable interaction model was only applied to covariates that showed significant interaction effects in the unadjusted model. The adjusted relative risk ratio (ARRR) by gender was derived using contrasts.

## Results

### Sample characteristics

Of 4,866 adolescents invited, 1,229 (response rate = 25.3%) provided parental consent and assent and completed the survey. The majority of the survey respondents were female (63.6%) and from public high schools (68.4%). While having a similar split across age, nearly a third were from Grades 9 (28.5%) and 10 (29.5%). Among the participants, 14.4% described the economic status of their family as not (particularly) as well off as other families, and 8.5% also reported that they currently do not live with any of their biological parents. SDQ assessment showed an average total score of 16.7. Internalising, externalising and prosocial mean scores were 8.9, 7.7 and 7.6, respectively (refer to Table 1).

### Suicidal ideation, suicidal attempt and self-harm

Overall, 54.0% (*n* = 664) of the sample reported having thoughts of attempting suicide since the onset of the COVID-19 pandemic (i.e., since 30 January 2020). In comparison, 24.2% (*n* = 297) had attempted suicide, and 34.2% (*n* = 420) had experienced self-harm (see Figure 1). Among all participants, 18.1% (*n* = 223) reported experiencing all three behaviours: suicidal ideation, suicidal attempt and self-harm, while 16.1% (*n* = 198) endorsed two of these behaviours. Of the 420 who reported self-harm, 53.8% (*n* = 226) had also reported attempting suicide, and 83.3% (*n* = 350) with either suicidal behaviour. This is equivalent to 18.6% and 28.4% of the total sample, respectively.

As illustrated in Figure 2, suicidal ideation, suicidal attempt and self-harm were further analysed by grouping adolescents: (1) those who did not experience suicidal ideation, suicidal attempt or self-harm (*n* = 492, 40.0%, 95% confidence interval [CI] = 37.3–42.8); (2) those who experienced suicidal ideation and/or self-harm but not suicide attempt (*n* = 440, 35.8%, 95% CI = 33.2–38.5); and (3) those who experienced suicidal attempt (*n* = 297, 24.2%, 95% CI = 21.2–26.6) regardless of having comorbid suicidal ideation or self-harm.

**Table 1.** Sociodemographic characteristics and social and emotional well-being of the study sample

| Key characteristics | |
|---|---|
| **Gender [*n* (%)]** | |
| Female | 781 (63.6%) |
| Male | 428 (34.8%) |
| Other | 20 (1.6%) |
| **Age in years [*n* (%)]** | |
| 13 | 300 (24.4%) |
| 14 | 351 (28.6%) |
| 15 | 315 (25.6%) |
| 16 | 263 (21.4%) |
| **Grade [*n* (%)]** | |
| 7 | 92 (7.5%) |
| 8 | 306 (24.9%) |
| 9 | 350 (28.5%) |
| 10 | 362 (29.5%) |
| 11 | 119 (9.7%) |
| **Type of school [*n* (%)]** | |
| Public | 841 (68.4%) |
| Private | 388 (31.6%) |
| **Perceived socio-economic status [*n* (%)]** | |
| Not well/Not particularly well | 177 (14.4%) |
| Fairly well | 731 (59.5%) |
| Rather well/Very well | 321 (26.1%) |
| **Living arrangement [*n* (%)]** | |
| With both biological parents | 874 (71.1%) |
| With either the biological mother or father | 251 (20.4%) |
| Does not live with any biological parent | 104 (8.5%) |
| **Strengths and Difficulties Questionnaire score [$\bar{x}$ (IQR)]** | |
| Total score | 16.7 (13.0–20.0) |
| Internalising score | 8.9 (6.0–11.0) |
| Externalising score | 7.7 (6.0–10.0) |
| Prosocial score | 7.6 (6.0–9.0) |

By gender, more females reported suicide attempts (29.6%, 95% CI = 26.5–32.9) and suicidal ideation or self-harm (39.6%, 95% CI = 36.2–43.0) since the pandemic, compared to males (13.1%, 95% CI 10.2–16.6 and 29.2%, 95% CI = 25.1–33.7, respectively). Of the 20 who identified as other gender, 50.0% (95% CI = 28.2–71.8) reported having attempted suicide, while 30.0% (95% CI = 13.4–54.3) reported experiencing suicidal ideation or engaging in self-harm.

### Regression analyses

Gender differences were confirmed by multinomial logistic regression. Compared to males, females were three times more likely to report a suicide attempt (ARRR = 2.97, 95% CI = 2.04–4.32) and two times more likely to experience self-harm or suicidal ideation (ARRR = 2.05, 95% CI = 1.48–2.83) since the pandemic (refer to

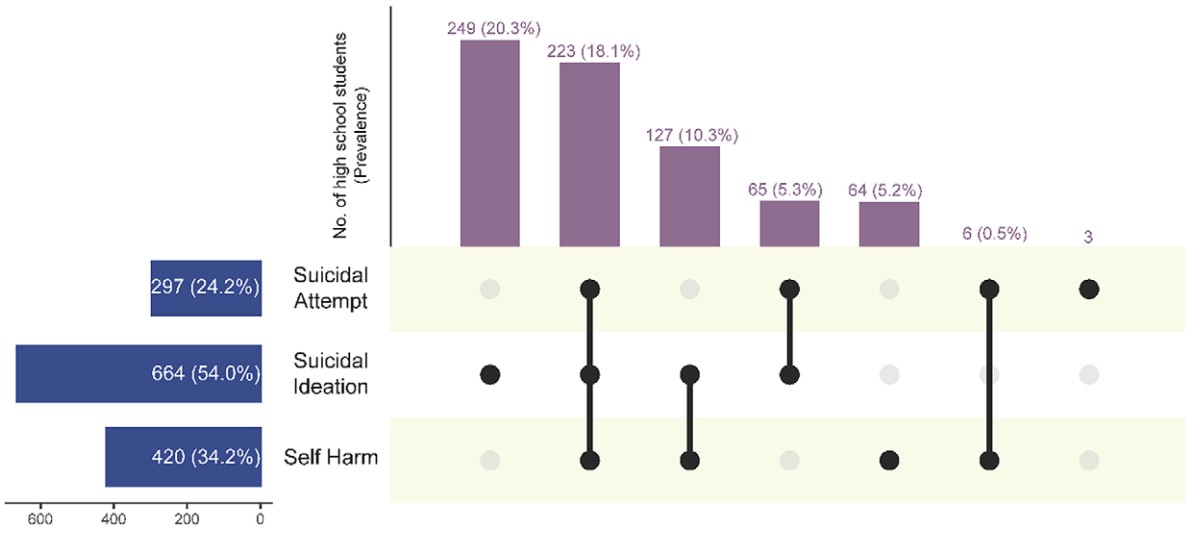

**Figure 1.** Prevalence of suicidal behaviours and self-harm since the COVID-19 pandemic among Filipino high school students. *Note*: Vertical bars (numbers indicate the number of students who experienced respective combinations of suicidal behaviours and self-harm; the percentage indicates the prevalence of respective combinations of the total sample; the number and percentage of those who did not experience any behaviours were not presented). Horizontal bars (numbers indicate the number of students who experienced the specific behaviour presented; the percentage indicates the prevalence of the specific behaviour of the total sample and percentages do not sum up to 100% as students may have experienced multiple behaviours).

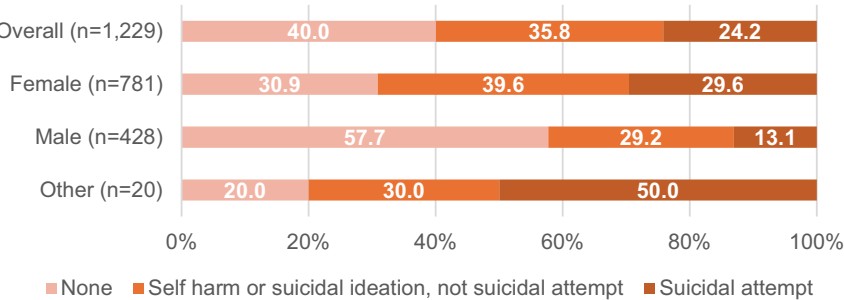

**Figure 2.** Prevalence of self-harm, suicidal ideation and suicidal attempt since the COVID-19 pandemic among Filipino high school students, by gender. *Note*: None (adolescents who did not report self-harm, suicidal ideation or suicidal attempt since the COVID-19 pandemic). Self-harm or suicidal ideation, not suicidal attempt (adolescents who reported self-harm or suicidal ideation, but not suicidal attempt since the pandemic). Suicidal attempt (adolescents who reported suicidal attempt since the pandemic, regardless of self-harm or suicidal ideation behaviours).

Table 2). Those who identified as the other gender showed an ARRR of 8.99 (95% CI = 2.62–30.92) for suicide attempt.

Adolescents who perceived their family as not economically well-off showed a higher prevalence of suicide attempts (ARRR = 1.91, 95% CI = 1.17–3.13) since the pandemic compared to those who reported an economically well-off family. No significant associations were detected between age and type of school and suicidal ideation, suicidal attempt and self-harm. Those who do not currently live with any of their biological parents were more likely to experience suicide attempts (ARRR = 2.93, 95% CI = 1.57–5.47) and self-harm or suicidal ideation (ARRR = 2.00, 95% CI = 1.19–3.37) when compared to those living with both biological parents. No statistical difference by gender was found for this covariate, despite the consistently significant risk associations among females and males (refer to Table 2).

The association between living with either of the biological parents and self-harm or suicidal ideation was moderated by gender (*p* < 0.001) (refer to Table 3). Among females, those who currently live with any of their biological parents were more likely to report

self-harm or suicidal ideation (ARRR = 1.63, 95% CI = 1.06–2.51) in reference to those who live with both biological parents. On the other hand, this association was reversed among male adolescents, where those who live with either of the biological parents were less likely to experience self-harm or suicidal ideation since the pandemic (ARRR = 0.52, 95% CI = 0.41–0.67) compared to those living with both parents. Effect modification by gender was also noted for suicidal attempts (*p* = 0.001). Adolescents who live with either parent showed an ARRR of 2.28 (95% CI = 1.29–4.03) among females and an ARRR of 0.63 (95% CI = 0.29–1.35) among males, which were found statistically different based on the interaction test.

## Discussion

This study revealed a high prevalence of suicidal ideation, suicidal attempt and self-harm among young Filipino adolescents. We found that 24.2% of adolescents in school have attempted suicide since the

**Table 2.** Adjusted relative risk ratio (ARRR) of correlates of self-harm, suicidal ideation and suicidal attempt since the COVID-19 pandemic among Filipino high school students

| | Self-harm or suicidal ideation, not suicidal attempt | Suicidal attempt |
|---|---|---|
| | ARRR (95% CI) | ARRR (95% CI) |
| **Gender** (*ref. Male*) | | |
| Female | **2.05 (1.48–2.83)*** | **2.97 (2.04–4.32)*** |
| Other | 2.94 (0.99–8.75) | **8.99 (2.62–30.92)*** |
| **Age** | 0.90 (0.73–1.11) | 0.98 (0.80–1.21) |
| **Perceived family economic status** (*ref. Rather well/Very well*) | | |
| Not well/Not particularly well | 0.93 (0.52–1.63) | **1.91 (1.17–3.13)*** |
| Fairly well | 1.00 (0.69–1.46) | 0.93 (0.59–1.47) |
| **Living arrangement** (*ref. with both biological parents*) | | |
| With either the biological mother or father | 0.99 (0.77–1.26) | 1.41 (0.89–2.22) |
| Does not live with any biological parent | **2.00 (1.19–3.37)*** | **2.93 (1.57–5.47)*** |
| **Type of school** (*ref. Public*) | | |
| Private | 1.27 (0.92–1.75) | 1.15 (0.84–1.59) |
| **SDQ score** | | |
| Internalising score | **1.15 (1.09–1.22)*** | **1.26 (1.18–1.33)*** |
| Externalising score | **1.18 (1.12–1.25)*** | **1.30 (1.22–1.39)*** |
| Prosocial score | 1.03 (0.96–1.12) | 1.02 (0.92–1.12) |

*Note*: Base outcome is None (adolescents who did not report self-harm, suicidal ideation or suicidal attempt since the COVID-19 pandemic).
Self-harm or suicidal ideation, not suicidal attempt (adolescents who reported self-harm or suicidal ideation, but not suicidal attempt since the pandemic).
Suicidal attempt (adolescents who reported suicidal attempt since the pandemic, regardless of self-harm or suicidal ideation behaviours).
Adjusted for gender, age, perceived family economic status, biological parents living with the adolescent, type of school, and SDQ scores; *p<0.05, **p<0.01; ***p<0.001.

**Table 3.** Effect modification of living arrangement by gender on self-harm, suicidal ideation and suicidal attempt since the COVID-19 pandemic among Filipino high school students

| | Females | Males | Interaction test |
|---|---|---|---|
| | ARRR (95% CI) | ARRR (95% CI) | *p*-value |
| **Self-harm or suicidal ideation, not suicidal attempt** | | | |
| Living arrangement (*ref. with both biological parents*) | | | |
| With either the biological mother or father | **1.63 (1.06–2.51)*** | **0.52 (0.41–0.67)*** | **<0.001** |
| Does not live with any biological parent | 2.27 (0.99–5.21) | 1.52 (0.67–3.45) | 0.558 |
| **Suicidal attempt** | | | |
| Living arrangement (*ref. with both biological parents*) | | | |
| With either the biological mother or father | **2.28 (1.29–4.03)*** | 0.63 (0.29–1.35) | **0.001** |
| Does not live with any biological parent | **3.02 (1.27–7.16)*** | **3.34 (1.48–7.53)*** | 0.871 |

*Note*: Base outcome is None (adolescents who did not report self-harm, suicidal ideation or suicidal attempt since the COVID-19 pandemic).
Self-harm or suicidal ideation, not suicidal attempt (adolescents who reported self-harm or suicidal ideation, but not suicidal attempt since the pandemic).
Suicidal attempt (adolescents who reported suicidal attempt since the pandemic, regardless of self-harm or suicidal ideation behaviours).
Participants who reported "Other" gender were excluded due to small cell size.
Adjusted for age, perceived family economic status, biological parents living with the adolescent, type of school, and SDQ scores; *p<0.05, **p<0.01; ***p<0.001.

pandemic, while 35.8% experienced suicidal ideation and/or self-harm. Adolescent girls reported a higher prevalence of suicidal ideation, suicidal attempt and self-harm compared to adolescent boys. This research also found an increased likelihood of suicidal attempts among those with poor socio-economic conditions and those not living with any biological parent. This study pioneered the collection of data on internalising and externalising

psychopathologies using a robust instrument, such as SDQ, at the tail-end of the pandemic and established its associations with suicide behaviours and self-harm in the Philippines (Zhang et al., 2025).

The prevalence of adolescents with suicidal attempts in our study is consistent with national and regional pre-pandemic estimates from the 2019 GSHS in the Philippines (Philippines Department of Health, 2023). Our analyses of its microdata revealed a

national past 12-month prevalence of suicide attempts of 24.3% (95% CI = 22.7–25.8) among 13- to 16-year-olds; the prevalence in Luzon, where the study sites are located, showed 23.9% (95% CI = 21.7–26.2). On the other hand, national and regional estimates based on the 2021 Young Adult Fertility and Sexuality Study (YAFS) in the Philippines showed a lower lifetime prevalence of suicide attempts among 15- to 19-year-olds during the pandemic (University of the Philippines Population Institute 2023a; b). In 2021 YAFS, the regional prevalence of suicide attempt in the CALABARZON region (6.3%), where our study site is located (i.e., the Province of Cavite), was lower than the national average (7.5%), as well as the three regions with the highest prevalence: the National Capital Region (13.9%), Bicol (8.6%) and MIMAROPA (8.4%), despite being geographically adjacent to CALABARZON (University of the Philippines Population Institute, 2023a). The 2021 Philippine National Survey for Mental Health and Well-being similarly showed lower lifetime prevalence of suicide attempts among adolescents in Grades 7–10 (17.2%) and those in Grades 11–12 (21.0%). The low prevalence of suicidal attempts in these recent national surveys may be explained by their interviewer-administered design compared to our study and the GSHS, which implemented a self-administered questionnaire. This reflects adolescents' hesitation to disclose such behaviour due to stigma (Erskine et al., 2024), particularly when surveys were conducted in households where parents or other adults were present or interviewed.

Our findings, along with data from the 2019 GSHS in the Philippines, align with previous studies showing no significant increase in suicidal behaviours and self-harm during the pandemic (John et al., 2021). Despite this stable trend, the prevalence of suicide attempts in the Philippines is notably higher than in neighbouring countries, such as Brunei Darussalam, Indonesia and Vietnam (Mahumud et al., 2022; Erskine et al., 2024). Moreover, the prevalence of suicide attempts observed in our study exceeds both global and regional lifetime and past-12-month pooled estimates reported in a meta-analysis of 316 studies (Van Meter et al., 2023). These between-country differences may be partly explained by the relatively late passage of national youth mental health legislation in the Philippines (Alibudbud, 2023; Mudunna et al., 2025; Szücs et al., 2025). For instance, Malaysia introduced its Mental Health Act in 2001, compared to the Philippines, which only enacted its Mental Health Act in 2018 and continues to face challenges, such as a low mental health workforce-to-population ratio of 1.68 workers per 100,000 population, compared to 5.86 in Malaysia (World Health Organization, 2020). Another possible explanation is the Philippines' relatively advanced integration of mental health literacy into school curricula (Shibuya et al., 2025). Mental health education is embedded in subjects such as health, values education and homeroom guidance, whereas countries like Indonesia incorporate it primarily through religious education (Shibuya et al., 2025). This broader curricular approach may have increased mental health awareness and literacy among Filipino adolescents, potentially contributing to greater self-reporting of suicidal behaviours and self-harm.

Suicidal ideation and suicide attempts were commonly experienced by adolescents with self-harm. A systematic review of 64 studies suggested the co-occurrence of suicidal behaviours with self-harm noting that self-harm usually precedes suicidal behaviours (Grandclerc et al., 2016). A grounded theory study of Filipino adolescents from the same site as this study found that self-harming adolescents experienced feelings of emptiness and powerlessness (Masana et al., 2021). While self-harm is considered a coping

mechanism separate from suicidal behaviours, the co-occurrence of these mental health outcomes indicates that Filipino adolescents are at risk of self-destructive behaviours. School-based and clinical samples in Portugal revealed a 38.1% lifetime suicide attempt among those who had lifetime self-harm, which suggested the precedence of self-harm provides an opportunity for early intervention in preventing suicidal attempts among self-harming youth (Duarte et al., 2020). It is also important to note that self-harm and suicide attempts are commonly difficult to ascertain because an individual's intent may not be clear, as motivations can shift within an episode and across episodes (Kapur et al., 2013). This challenge may be more pronounced in situations involving multiple motivations for self-injury and among younger adolescents (Kapur et al., 2013). This complexity is reflected in our findings, where self-harm was reported alongside suicidal attempt by 2 in 10 adolescents and alongside suicidal ideation by three in ten adolescents (refer to Figure 1).

Our study demonstrated correlations between self-harm, suicidal ideation and suicidal attempts, as well as internalising and externalising SDQ scores. Mental health problems usually emerge during adolescence (McGrath et al., 2023), reflecting neurodevelopmental vulnerabilities of young people at this critical stage of psychosocial development (Australian Institute of Family Studies, 2017; Duarte et al., 2020; Erskine et al., 2024). This highlights the importance of screening and basic psychosocial support to address common psychopathologies as a preventative strategy and building capacities among the specialist and non-specialist workforce, such as teachers and school counsellors (Alibudbud, 2023; World Health Organization Regional Office for the Western Pacific, 2023).

Gender differences were observed in this study, as females were at least two times more likely to report self-harm or suicidal ideation (OR = 2.05) and suicidal attempts (OR = 2.97) compared to males. This is consistent with other surveys, which found an increased risk among females. A global GSHS study revealed significantly higher odds of suicide attempts among females (OR = 1.45), and a global meta-analysis found a significant prevalence difference between females and male adolescents (8.5% vs. 4.9%) (Van Meter et al., 2023). The association between gender and suicidal behaviours and self-harm can stem from known gender differences in affective psychopathologies and risk factors (Claes et al., 2007; Turecki and Brent, 2016; Orri et al., 2020).

This study found an increased risk of self-harm, suicidal ideation and suicidal attempts among adolescents who do not live with any of their biological parents. These adolescents in our sample reside with their grandparents (4.2%), other relatives such as older siblings, uncles/aunts (3.4%) and other people (1.1%). This aligns with a national survey in the Philippines conducted during the pandemic, which found the highest prevalence of lifetime suicide attempts among children and adolescents who were living with their relatives other than their mother or father (De Guzman et al., 2025). These findings are consistent with studies conducted in East Asia, Scandinavia and Sub-Saharan Africa, which have demonstrated significant associations between parental absence and poor mental health outcomes (Nrugham et al., 2010; Samm et al., 2010; Noh, 2019; Zhang et al., 2019; Annor et al., 2024). Parental absence is widely recognised as an indicator of childhood adversity, often linked to unstable home environment and non-secure attachments. These conditions may impair emotional regulation and increase vulnerability to high-risk behaviours, including self-harm and suicidal tendencies (Annor et al., 2024). Notably, parental absence other than orphanhood has been found to uniquely contribute to poor adolescent mental health, independent of other childhood

adversities (Annor et al., 2024). This underscores the importance of parental support and healthy parent-child relationships, particularly in cultural contexts like the Philippines, where strong social expectations favour intact family structures. A qualitative study among self-harming Filipino adolescents highlighted how the presence of parents at home created opportunities for adolescents to share their thoughts and emotions, while their absence or perceived unavailability hindered such communication (Masana et al., 2021).

During the COVID-19 pandemic, when adolescents' social environments were disrupted, parents became the primary source of social support not just in the Philippines but also in other countries such as Indonesia and Vietnam (UNICEF et al., 2022; Maravilla et al., 2025). This support was found to have a protective effect on youth mental health, reducing the risk of suicidal ideation, self-harm and any mental disorder (Maravilla et al., 2025). However, parents in the Western Pacific Region were reported to have the lowest level of parental understanding and monitoring compared to other global regions (Kushal et al., 2021). Strengthening parental mental health literacy and parenting efficacy is therefore essential to empower families in supporting adolescents experiencing mental health problems.

The relationship between living arrangements, suicidal ideation, suicidal attempt and self-harm was moderated by gender. Among girls, living with one biological parent increased the odds of suicide attempt, as well as suicidal behaviours and/or self-harm, compared to those living with both parents. However, an inverse association was observed among boys, that is, those living with one biological parent showed a lower risk for suicidal behaviours and self-harm in reference to adolescent boys living with both parents. While the detrimental relationship of not living with either parent with youth mental health remained consistent between genders, girls from single-parent families showed a higher prevalence of suicidal behaviours and self-harm. A meta-analytic review also revealed the increased prevalence of self-harm among single-parent families compared to those with both parents in the household (OR = 1.20), although this review did not report gender-specific associations (Xiao et al., 2022). A study from China did not find a significant interaction between gender and family structure (Zhang et al., 2019). Gender perspectives on kinship care are not well studied in adolescent mental health, which opens research opportunities for societies with evolving social values and gender norms (Kan and Zhou, 2022).

Living circumstances, such as parental separation and poor socio-economic conditions, are usually beyond the control of adolescents, leading to their feelings of powerlessness (Stanley et al., 2012). It is key to empower young people in these situations to enhance their confidence, self-worth, self-awareness and help-seeking behaviours. Boosting young people's resilience may also improve their emotional regulation and avert potential risks for suicidality and self-harm.

## Limitations

This study has limitations. First, the results from our survey are not nationally representative. Despite not being a representative survey, our study revealed an elevated prevalence of suicidal attempts among 13- to 16-year-old Filipino adolescents, consistent with previous national surveys in the Philippines. Our findings warrant further investigation through a large-scale survey to explore patterns in psychopathologies among young adolescents in the post-pandemic period. Second, our low response rate may indicate bias, as only one in every four parents had given their consent for their adolescent to participate. To note, all adolescents with parental consent provided assent. Third, the fixed reference time point (i.e., 30 January 2020) used for the recall period in this study limits its comparability with existing studies. Given that the onset of suicidal behaviours occurs after the age of 10 years (Voss et al., 2019), our study estimates may reasonably be interpreted as lifetime prevalence. A repeated national youth survey in the Philippines (2002–2021) consistently found suicide attempt onset within 0–2 years before their survey interview, with the recent survey indicating the highest incidence in 2020 (University of the Philippines Population Institute, 2023a). Although recall periods differ between our study and prior research, a global meta-analysis of 365 studies found no strong evidence of statistical differences between lifetime and past 12-month prevalence of suicidal ideation and attempt among young people (Van Meter et al., 2023). Lastly, associations between suicidal behaviours, self-harm, psychopathologies and socio-demographic characteristics found by our study should be interpreted with caution and should not be considered causal due to the cross-sectional nature of our survey.

## Conclusions

This study revealed a high post-pandemic prevalence of self-harm, suicidal ideation and suicide attempts among young adolescents in the Philippines. These findings underscore the urgent need for policymakers and stakeholders to take immediate action, especially in light of the slow-paced health reforms related to suicide prevention in the country. This decade-long health crisis affecting younger Filipino adolescents demands a coordinated and evidence-based response.

Beyond gender disparities, the study also uncovered intersecting vulnerabilities involving gender and living arrangements with parents. These insights are critical for informing both clinical interventions and public health strategies with the rapidly evolving social norms and environments in the post-pandemic context. The consistent associations between suicidal behaviours and common psychopathologies highlight the importance of early screening and early intervention and the need to strengthen mental health workforce capacity in schools and communities.

**Open peer review.** To view the open peer review materials for this article, please visit http://doi.org/10.1017/gmh.2025.10105.

**Supplementary material.** The supplementary material for this article can be found at http://doi.org/10.1017/gmh.2025.10105.

**Data availability statement.** The data that support the findings of this study are available from the lead author, DHA, or the corresponding author, JCM, upon reasonable request.

**Acknowledgements.** The authors would like to thank Dr Sonja Gilbert for her contributions to the design and planning of the study; Jhoanna Peña, Maricar Comabras and Rosanna Espinosa for assisting with data collection and coordinating data encoding; and Armen Jheannie Barrameda for supporting the team with data cleaning. Importantly, the authors would like to acknowledge the support of schools, parents and students for their support and participation in the study.

**Author contribution.** DHA, RCT, JPG, ASR, IP and JCM conceptualised the research question and study design. DHA, RCT, JPG, ASR and IP collected the data. JCM conducted all statistical analyses. DHA and JCM wrote the first draft of the manuscript. All authors contributed to the revised manuscript. All authors approved the submitted version of the manuscript.

**Financial support.** This study was supported financially by the De La Salle Medical and Health Sciences Institute, Philippines, and the University of Turku, Finland. JCM is supported by the Centre of Research Excellence in Adolescent Health (APP1171981), which receives funding from the National Health and Medical Research Council in Australia.

**Competing interests.** The authors declare none.

**Ethics statement.** This study is approved by the De La Salle Medical and Health Sciences Institute Independent Ethics Committee (GSER-038). This committee operates in accordance with the Declaration of Helsinki, National Ethical Guidelines for Health Research, Council for International Organizations of Medical Sciences and the International Conference on Harmonization/Good Clinical Practice. Informed consent was obtained from parents and students.

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
