## [Reviewer Report]

This manuscript investigates suicidal ideation, suicide attempts, and self-harm among Filipino high school students during the pandemic. It draws on a large sample size, uses the Strengths and Difficulties Questionnaire (SDQ), and applies multinomial logistic regression. The topic is timely and significant, with potential to inform mental health interventions.

1. Clarify “Suicidal Behaviours” as a Concept

The manuscript uses the term “suicidal behaviours and self-harm” (e.g., p.7, L 52), but would benefit from clearly defining which acts are included under “suicidal behaviours.”

Since self-harm, suicidal ideation and suicide attempt are thoughtfully treated as distinct variables in the analysis, providing a concise operational definition of the term “suicidal behaviours” would further enhance conceptual clarity and consistency across the manuscript.

A one-sentence definition in the Introduction or Measures section would suffice.

2. Table and Figure Labeling

Table 2 and 3 list “suicidal behaviours and self-harm” in the titles, but the column headings refer to distinct categories :

• Self-harm or suicidal ideation, not suicidal attempt

• Suicidal attempt

While the categorization aligns with the multinomial regression model, adding clearer labels or brief footnotes could help readers more easily grasp how the categories are defined.

3. Questionnaire Sequencing and Language Clarification

Consider describing the general questionnaire (and its Filipino translation) first, followed by the English-only SDQ.

4. Terminology Sensitivity: Avoiding “Committing Suicide”

It’s commendable that the manuscript now uses the inclusive term “attempted suicide.” Nonetheless, the original English item referred to “committing suicide.” Because many bodies – including the International Association for Suicide Prevention (IASP) and WHO – discourage this wording due to its historical links with criminality and stigma, you might wish to add one brief sentence explaining how the Filipino translation rendered that phrase. Such a note would reassure readers about the study’s ethical and culturally sensitive approach and offer guidance for future survey design.

5. Minor Consistency Gaps

Page 14, (L 30-31) mentions “suicidal attempts and self-harm”, omitting ideation

Page 15, (L 11), includes “self-harm or suicidal ideation, and suicide attempt.”

Consistency across sections will ensure clarity for the reader.

Overall, these minor refinements will make an already valuable manuscript even clearer for an international readership.

---

## [Reviewer Report]

Overall comments:

Citation style needs to check, because it does not follow any specific reference style. For example, … 15-24 year-olds (Maligalig 2021). Here, a comma is missing. I found the authors followed this style. Additionally, page no is confusing because every page has different page numbers. Please note that I followed the “page 1 out of 26” format, which is a top left side header. Please use continuous line number instead of restarting each page.

However, the methodology and discussion sections need more attention. I have provided my specific reviews. Currently, this manuscript is not ready for publication, it needs major modification. I hope, authors will address the comments which will strengthen the quality of this manuscript.

Impact statement: “To our knowledge” is not scientific language, we need empirical evidence. Whether any research is the first or last does not convey any significance to literature or application, the impact matters. Hence, focus on the impact.

Introduction:

In line 5 (page 4), self-harm without intention of suicidal behavior falls under non-suicidal self-jury; hence, use one term or write: non-suicidal self-injury (e.g., self-harm).

For line 23-26 (page 4), it would be better if you cite against your claim that data limitation in low- and middle-income countries. Literature review suggests there are significant amount of research about suicidal behavior conducted in these countries (E.g., India, Nepal, Bangladesh, and African Countries).

For line 23-54 (page 5), as a reader, I did not grasp the key point from this paragraph. Could you please revamp the writing coherently? This paragraph represents several ideas; for example, data for high income countries, lack of data in low- and middle-income countries, introduction of pandemic (please mention whether it is covid or other pandemic), psychological disorders among children, and research gap. It would be better if you convey the message using several paragraphs.

For 8-10 (page-5), please include geographical location of YAFS. In line 16 (page-5), what YAFS did not include is missing. Please specify it.

For line 36-37 (page 5), please specify the pandemic year, age range of adolescents. In line 45, prosocial behavior term suddenly popped up, it is better to explain it in the earlier paragraph because we do not know anything about it and how this topic is aligned with this study. For this paragraph, please follow the same tense form.

Note: Could you please include one more paragraph, which can explain the research significance of this study?

Method:

Design and sample:

It is good at describing the geographical location. Could you include why you have selected these two cities?

For 32-55 (page 6), Thank you for explaining the consent-taking procedures. I am ok with that, but my prime concern is ethical clearance from an established ethics committee. It is mandatory because the sample is human beings. For this research, the population is adolescent, and they are considered vulnerable. The risk of vulnerability increases by adding suicidal behavior aspect in this study. I believe you have obtained ethical clearance from an ethics committee. Please include it in this section.

Could you please depict the following concerns:

1. How did you maintain the confidentiality of the data?

2. As this subtopic represents sample, please explain the sampling techniques and sample size.

Procedures:

1. Who were the research team members? Are they the same people mentioned as authors? If not, did they receive training on data collection and ethics?

2. For 20-22, you explained that teachers were present during the administration of survey questions to support in times of distress. I assume you are talking about psychological distress. In this case, please write about whether these teachers had training on dealing with psychological distress or not.

3. In line 37, you stated that participants who experienced distress were referred to student counsellors. Please explain how your team identified distressed participants. Were they distressed during or after the survey?

4. As this research was sensitive in nature, did you conduct any pilot study to modify and validate the survey questionnaires?

5. The duration of the administration of the survey questionnaire is not stated. Please mention it.

Study Measures

The authors stated that the “questionnaire is composed of seven modules and 125 items outline in Table S1”. I have reviewed this table, but I did not find any questionnaire that consists of seven modules and 125 items.

For self-harm question, “intentionally” word is misleading. Self-harm is an intentional act, which may occur with or without suicidal intent. Could you please clarify it?

For socio-emotional well-being: Please explain about the SDQ scale with appropriate citations; for example, who developed it, with whom this scale can be administer, usage of the scale, psychometric properties of this scale etc.

Results

For line 21 (page 10), Please use the specific time instead of since the pandemic.

For line 24 (page 10), the writings suggest that 24.2% or 34.2% participants had suicidal attempts and self-harm. Do the values 54%, 24.2%, and 34.2% indicate number (as percentage) of participants? If yes, please rewrite it because a fraction number or a percentage does not represent human beings.

Discussion

For line 18-22, I understand that this study is the first study in Philippines, it is ideal to support and discuss your findings critically in the discussion section.

For line 25-42 (page 13), it is good that you have compared the findings with previous studies conducted in Philippines. The comparison suggests a consistent result which was conducted in pre-pandemic situations. Is it possible to explain the reason why the prevalence of suicidal attempts is similar, though research evidence indicates that suicidal behavior increased during and after pandemic. You can also strengthen your claim including global studies which have similar cultural and economic contexts.

For line 45 (page 13)-27 (page 14), this paragraph is quite a large paragraph and contained several points of discussion. I suggest breaking them into specific topics. Please include the critical discussion along with data comparison; for example, what possible reasons can explain this discrepancy or similarity of the findings.

Limitations: In the third limitation, you have mentioned recall period, which is a new concept you have introduced in the limitation section. What is recall period? If this concept is aligned with methodology, please introduce it there first.

In the last limitation, it is stated that associations found. Please precisely articulate associated factor; for example, association between A and B.

Thank you.

---

## [Reviewer Report]

Suicidal behaviours and self-harm among adolescents: Results from a school-based mental health survey in the Philippines

Thank you for the opportunity to review this important study, which provides prevalence estimates for youth suicide risk in the Philippines - a LMIC setting underrepresented in the scientific literature. The methodology, procedure, and data analysis are a unique contribution to the global literature, as there remains some debate on how to evaluate self-harm risk along with more traditional screening for suicide ideation and attempt. Further, the research manuscript achieves large data collection goals to understand a relevant and current global problem – youth suicide thoughts and behaviors.

Below please find comments and questions regarding the manuscript presentation:

1. The authors use the words “children”, “adolescents”, and “young people” to refer to the student sample. Please choose one term and use throughout the paper when referring to the students.

2. Is the following sentence complete? “For example, a recent global review of evidence on self-harm found studies in high-income countries (HICs) in North America, Europe and Asia such as China, Taiwan and South Korea (Moloney et al. 2024).” Maybe authors meant to say that the global review primarily focused on high income countries.

3. Re-word the sentence on pg. 3, line 12 to read: A recent study summarizing results from LMICs used XXXX-2019 data from the Global School Health Survey (GSHS). There remains a need for LMIC studies, since 2019, that estimate and seek to understand suicide behaviors among students.

4. Run-on sentence. Pg 3, line 34. Consider, “Occasional school closures, post-pandemic, challenged students’ social skills, life routines, and management of pre-existing mental health symptoms.”

5. Re-word the sentence on pg 3, line 40 to read “ Given disparate and underrepresented research studies from low-resource contexts, stakeholders and local governments are in need of information to enhance policies and supportive actions for students’ well-being.

6. ASEAN acronym is not previously defined.

7. Re-word sentence on pg 4, line 12. “According to the Young Adult Fertility and Sexuality Survey (YAFS) suicide attempts among 15-24 year olds increased from 3% in 2002 to 7.5% in 2021”

8. Are suicide attempts increasing among 15-24 year olds, but suicide mortality decreasing in this same group? The Maligalig 2021 citation about decreasing mortality in 15-24 year olds seems to contradict the previous sentence stating that suicide attempt have increased for 15-24 year olds (Univ of Philippines, 2022). Please clarify.

9. In scientific writing present the IV and DV in each sentence. I would edit the below statement to read: The majority of research from not just in the Philippines and but also in other LMICs has focused on cross-sectional associations between internalising symptoms and INSERT OUTCOME VARIABLE (SUICIDE ATTEMPT?)

10. In scientific writing present the IV and DV in each sentence. Re-word pg 4, line 21, “Existing studies conducted in the Philippines did not account for externalizing symptoms in models that tested associations to suicide ideation? Behaviors?.”

11. Consider re-wording. A cross-sectional survey was implemented in public and private high schools located in two select cities in Cavite Province, Philippines: Dasmarinas City and Trece Martirez City.

12. Cavite is the most populous province in the Philippines and has undergone rapid industrialization resulting in a to diverse population of migrants or relocators and relocation of informal settlers from the National Capital Region of the country (Mapa 2021; Provincial Government of Cavite 2022).

13. The authors compare 16.18% population in Dasmarinas to 210,504 population in Trece Martirez. Which metric, percent or whole numbers will be used for comparison? Choose one. Percentages are preferred. Also, the number of private and public schools are given for Dasmarinas, but not Trece Martirez. How many schools are in Trece Martirez? Be consistent.

14. Shorten sentence pg 5, line 26. Consent from parents was obtained by sending them an invitation letter that described the purpose of the study.

15. How many days prior to the survey administration were information sheets sent out? Be specific.

16. When were information sessions conducted? (Before or after letters were sent)

17. Did you receive 1229 consent from parents?

18. Run-on sentence, page 6 lines 1 and 2. Shorten.

19. Was a 25.3% response rate expected?

20. How were consenting and non-consenting students separated for survey administration? How many students were given the survey at a time?

21. How often or frequent was classrooms set up for survey administration? How many students on average were in each classroom when taking the survey?

22. About how long did it take for students to complete the survey questionnaires and study protocol?

23. How many students experienced distress and needed referral?

24. Provided detail about the SDQ subscale that will help the reader to interpret the score, range of scores and meaning of the subscale (eg. What are higher prosocial behaviour scores indicative of?). Revise sentence reporting SDQ mean scores on page 9 to add meaning/interpretation.

25. What are the “respective covariates”? Authors should name the covariates.

26. What does the author mean by “suicidal behaviours”? On page 8, it appears that suicidal behaviours mean suicide ideation and attempt. Yet, in other parts of the paper suicidal behaviours appears to mean self-harm and attempt. Clarify.

27. Is suicidal risk conceptualized on a continuum? Can authors add a sentence justifying select suicide categories (eg., suicide ideation and self-harm)? Manuscript results depict variation in suicide risk outcomes among students, I wonder how they conceptualize the different types of suicide thoughts and behaviors in their sample.

28. Add n’s in parentheses to percent results in 2nd paragraph of page 9 (under suicidal behaviour and self harm since the pandemic)

29. Why do the authors report, “of the 664 high school students who had suicidal ideation, only 249 did not report any suicide attempt…”? Are authors highlighting that the majority of students with suicide ideation also experienced attempt and self-harm behaviours?

30. Page 9, last paragraph, lines 27-36, the authors present a different order of suicide groupings than presented on page 8. Use parallel construction. For example if those who did not express suicide ideation, attempt, or self-harm was listed as number (1) on page 8, then this suicide group should also be listed as (1) on page 9.

31. On page 10, add the variable names of interest to this sentence: “Those with other gender showed an ARRR of 8.99.”

32. Present the comparison or control group when reporting moderated effects for students who lived with both parents versus those who lived with either of their biological parents (last paragraph page 10)

33. Summarize main results, using percents, at the beginning of the discussion.

34. Findings support the influence of parent/family relations and mental health symptoms on youth suicide outcomes. Suicide in Elementary School-Aged Children and Early Adolescents | Pediatrics | American Academy of Pediatrics

35. Add descriptive title to Table 1.

36. What are some plausible implications for policymakers and stakeholders in light of these findings? What kind of interventions are appropriate for suicide prevention in schools?

---

## [Editor Report]

Dear, Dr Alberto and colleagues,

Thank you for submitting your paper to the journal. It is very interesting and indeed helps address some of the data issues with LMICs in the pandemic. Please carefully read the reviews by our expert reviewers and address them per item. 

I have a few comments in addition to the reviews provided:

1. Please closely examine the language used e.g., line 12, ‘suicide is considered the third leading cause’, whereas it would imply that added consideration was given on top of existing data, whereas ‘suicide is the third leading cause of death’ reflects the process more accurately. Furthermore, these sentences ‘ For example, a recent global review of evidence on self-harm found studies in high-income countries (HICs) in North America, Europe and Asia such as China, Taiwan and South Korea (Moloney et al. 2024). The majority of data on suicidal behaviours from the Global School Health Survey (GSHS), which is a multicountry study in LMICs, was done in 2019 leaving a huge data gap in adolescent mental health post-pandemic (Li et al. 2021).’ may need to be revised for clarity as it is not immediately clear on reading. “With postpandemic data being skewed among HICs” is another example, with data being skewed “towards” HICs. For the benefit of the reader of this important work, I would suggest carefully examining the language prior to resubmission. 

2. Like the reviewers' comment, I would strongly suggest that the authors justify the choice in study location. Given that the background implied that this was a nationwide survey, these two choices would need to be justified in order to make conclusions and have implications on the country level. Otherwise, the rationale may need to be changed. On the same thread, a power analyses may need to be reported to see if the sample is representative of the population the authors wish to study. 

3. I would caution against using words such as ‘alarming’ as it would be up to the reader to conclude that. In order to raise this topic, I would suggest comparing it to other rates of self-harm and suicide globally. Having a more thorough discussion on how the Philippines would contextually differ, especially these locales, would strengthen the discussion section as now it focuses primarily on the link between the SDQ, parent-child relationships etc. In that vein, more information on these locations is needed. 

4. In the discussion section, on healthy parent-child relationships, I fully agree; however, it would need to be contextualised within the pandemic’s situation. 

I look forward to reading your resubmission.

All the best,

Dr. Sandersan Onie

---

## [Reviewer Report]

I thank the authors for their thoughtful and comprehensive revision of the manuscript. The revised version shows substantial improvement in clarity, methodological rigor, and overall readability. Key issues raised in the earlier review - such as the definition of suicidal behaviors, consistency of terminology, questionnaire sequencing, table and figure labeling, and inclusion of ethical details - have been fully addressed. The strengthened discussion and contextual comparisons further enhance the paper’s significance.

I am satisfied with the current version and recommend acceptance.

---

## [Editor Report]

Thank you for addressing the comments in a comprehensive manner. There are some edits that I would like to see addressed from the new text, including a reviewer comment that I feel could be better addressed by the authors:

1. In the abstract, you note the results since 30 January 2020; however, the end date is not mentioned. Similarly in the Impact Statement, the period of the study is not specified. 

2. On page 11 and 12, it was noted that All items except the SDQ were translated to Filipino and then back translated. However, it is also noted that no SDQ was available in Filipino. Please clarify why the team did not translate the SDQ if a Filipino version did not exist. 

3. Please separate study design and setting in the Methods section. 

4. When specifying the recall date, please also specify the date that the survey was conducted. 

5. As the reviewer has specified, given the high rate of response for suicide attempts and self-harm, please provide more information on these questions and how it is understood in Filipino. I found that this had not been responded to in the reviewer response table. Given that this is the major claim of the paper, and the core of the discussion, it would be great to expand on this. 

Thank you and all the best,

Dr Sandersan Onie

---

## [Editor Report]

Dear Prof Maravilla and colleagues,

Thank you for the revision and for addressing the comments provided. This expanded upon methodology section allows me to see the full picture of the manuscript and study. 

While it is substantially improved, I still have concerns surrounding comment 5 in my previous action letter, surrounding the questions on self-harm and suicide. Since the question is worded since a fixed time point, that is 30th of January 2020, and the response dates vary significantly from one another, it becomes very difficult to contextualise the findings in with other findings. While there is a point in the limitations about this, no statistical difference between past 12-month prevalence and lifetime prevalence does not suggest no difference, but lack of an evidence of one in that study. 

Furthermore, the self-harm and suicide attempt prevalence and rates can vary significantly depending on how the question is asked. Are there safeguards present to ensure uniform understanding of self-harm and suicide among respondents, and how does this format of data collection and question potentially affect its rates? For example, the discussion section cites the NAHMS study by Erskine and collegues, which were child-parent interviews. 

Furthermore, the discussion between self-harm and suicide attempt may need to be further tempered, given that intent is fluid, and it is difficult to ascertain the difference between a suicide attempt and self-harm. 

Another key comment is whether the region of the study has consistently high or low rates compared to other regions, to allow the reader to compare. 

All in all, this study has significantly improved, and I realise it has been through two rounds of review, but I would like to see the above points addressed prior to acceptance. Please address them thoughtfully and carefully. 

Thank you and all the best,

Dr. Sandersan Onie

---

## [Editor Report]

Dear Authors,

I am sufficiently satisfied with your responses to my queries and am recommending this article for publication.

Thank you and all the best,

Dr. Sandersan Onie